# Early Deaths in Childhood Cancer in Romania—A Single Institution Study

**DOI:** 10.3390/children8090814

**Published:** 2021-09-16

**Authors:** Doina Paula Pruteanu, Elena Diana Olteanu, Rodica Cosnarovici, Emilia Mihut, Radu Ecea, Nicolae Todor, Viorica Nagy

**Affiliations:** 1Department of Pediatric Oncology, “Prof. Dr. Ion Chiricuta” Oncology Institute, 400015 Cluj-Napoca, Romania; pauladpruteanu@gmail.com (D.P.P.); rodicacosnarovici@yahoo.com (R.C.); emiliami@yahoo.com (E.M.); radu.ecea@gmail.com (R.E.); todor@iocn.ro (N.T.); nagyviorica2012@yahoo.com (V.N.); 2Department of Radiation Oncology, Iuliu Hatieganu University of Medicine and Pharmacy, 400000 Cluj-Napoca, Romania

**Keywords:** childhood cancer, early deaths, middle income country, risk factors

## Abstract

(1) Background: Survival in childhood cancer has improved significantly over the last decades. However, early deaths (EDs) represent an important number of preventable deaths. Our aim was to provide more insight intoEDs in developing countries. (2) Methods: We conducted a retrospective analysis of patients aged 0–18 years with childhood cancer diagnosed between 1996 and 2008 and admitted in the Institute of Oncology “Prof. Dr. Ion Chiricuta” Cluj-Napoca (IOCN), Romania. After exclusion of patients (pts) older than 18 years at diagnosis, pts with a missing personal identification number and pts with unconfirmed diagnosis of malignancy, we included 783 pts in the final analysis. We defined ED as survival of less than one month after cancer diagnosis. We divided pts in groups according to age, major tumour categories and treatment time periods. (3) Results: ED was registered in 20 pts (2.55%). A total of 16EDs were registered in haematologic malignancies and 4 in solid tumours. Statistical analysis was performed on pts diagnosed with haematological malignancies. A statistically significant higher proportion of patients with performance status (PS) 3 and 4 died within one month after diagnosis (24.1%) than patients admitted with PS 0–2 (1%)—*p* < 0.01. We found no statistically significant difference regarding ED when comparing male versus female (*p* = 0.85), age at diagnosis or between the threeperiods of diagnosis (*p* = 0.7). (4) Conclusions: PS at admission is an important risk factor associated with ED in pts with haematologic malignancies. ED in our institution reflects frequent late presentation for medical care, late diagnosis and referral to specialised centres.

## 1. Introduction

Patients’ survival in childhood cancer has steadily improved over the last decades because of improved diagnosis and modern treatments [1,2]. However, there are patients who die shortly after diagnosis. Many of these patients are ineligible for enrolment in clinical trials or succumb before enrolment; hence, this group is poorly understood and investigated.

Several population-based studies conducted in developed countries have explored the issue of early deaths (EDs) in childhood cancer and identified the following risk factors: age younger than one year, acute myeloid leukaemia (AML) and hepatic tumours [3,4,5].

Disparity between treatment outcomes in developed countries versus developing and underdeveloped countries is well established in the literature, although in recent years differences have tended to diminish [6,7,8].

Our study aimed to provide more insight into ED and its associated risk factors in developing countries. To the best of our knowledge, this study is the first that addresses the issue of ED in Romania, a middle-income country that started using modern European protocols only in the mid-1990s.

## 2. Materials and Methods

We conducted a retrospective study of patients aged between 0 and 18 years with all childhood cancer diagnosed between 1996 and 2008 and admitted in our department at the Institute of Oncology “Prof. Dr. Ion Chiricuta” Cluj-Napoca (IOCN), Romania.

The study was approved by the Ethics Committee of Iuliu Hațieganu University of Medicine and Pharmacy.

Out of all patients admitted in IOCN between 1996 and 2008, 1389 were identified as potentially eligible for this retrospective analysis. We excluded 285 patients who received primary treatment outside IOCN (they received only the radiotherapy sequence or treatment for relapse in our Institute), 88 patients with incomplete or missing files, 166 patients older than 18 years at diagnosis, 65 patients with a missing personal identification number (unique identifying number) and 2 patients with an unconfirmed diagnosis of malignancy. After exclusion, 783 (56.4%) patients were included in the final analysis.

We defined early death (ED) as survival of less than one month after the diagnosis of cancer. Patients surviving one month or more after diagnosis was made were classified as non-early deaths.

Performance status was defined according to the Eastern Cooperative Oncology Group (ECOG) Scale.

The following information was extracted from patient files: demographic data, clinical data, oncologic diagnosis, acute toxicities, date of death and cause of death.

Comparison of categorical variables was performed using the chi-square (χ^2^) test. A *p* < 0.05 was considered statistically significant. The statistical analysis was performed with the STATA package (2013, STATA Statistical Software 13 ed; Stata Corporation, College Station, Texas, USA).

To facilitate the data analysis and to identify risk factors, we divided the patients into groups according to age (younger than one year, 1 to 4 years, 5 to 9 years, 10 to 14 years, 15 to 18 years), major tumour categories (haematologic malignancies, central nervous system (CNS) tumours and non-CNS solid tumours) and treatment time periods (1996–1999; 2000–2004; 2005–2008).

Staging information was analysed only for non-CNS solid tumours.

## 3. Results

A total of 783 patients younger than 18 years of age at diagnosis were included. ED was registered in 20 patients (2.55%), from a total number of deaths of 454, so the early death group represents 4.4% of all deaths in this period.

Clinical and demographic characteristics of children included in this analysis are presented in Table 1 and types of tumours are treated in Table 2.

Early deaths were reported in 16 patients diagnosed with haematologic malignancies and 4 with solid tumours of all the 20 EDs reported; in patients with CNS tumours, no EDs were registered.

Patient flow is presented in Figure 1.

### Causes of Early Death

Treatment abandonment, defined according to SIOP PODC (International Society of Paediatric Oncology Paediatric Oncology in Developing Countries), was registered in 58 patients (7 patients refused treatment and 51 patients abandoned treatment); however, no early deaths were reported in this group of patients

From the 20 EDs registered in this period, 19 deaths were related to disease progression, and 1 death was related to treatment (surgery). No ED related to relapse, misdiagnosis or failure to establish diagnosis was recorded in this cohort of patients.

Six patients diagnosed with acute lymphoblastic leukaemia (ALL) died within 30 days of diagnosis: three due to leukostasis and two due to brain haemorrhage and one due to septicaemia.

Patients diagnosed with acute myeloid leukaemia (AML) presented the following causes of death: brain haemorrhage (four patients), massive upper digestive tract haemorrhage (one patient).

Two early deaths were registered in patients with Burkitt’s lymphoma: one due to liver failure in the context of disease progression and one due to bowel obstruction.

In patients with non-Hodgkin non-Burkitt lymphoma, the following causes of ED were identified: liver failure due to tumour progression (one patient), respiratory failure due to tumour volume (one patient) and bowel obstruction (one patient).

Four EDs occurred in patients with solid tumours: a one-year-old boy diagnosed with stage IV sacrococcygeal germ cell tumour admitted with performance status (PS) of 3 died 12 days after diagnosis due to respiratory failure determined by disease progression; a two-year-old boy diagnosed with stage III parapharyngeal embryonal rhabdomyosarcoma admitted with PS of 4, died 5 days after diagnosis due to respiratory failure determined by disease progression; a one-year-old girl diagnosed with Stage III Wilms’ tumour admitted with PS of 3, died 5 days after diagnosis due to cardiac failure determined by disease progression; a 6-year-old girl diagnosed with Wilms’ tumour and cardiac malformations died 29 days after diagnosis due to postoperative complications.

Causes of death according to major cancer diagnostic groups are presented in Table 3.

In order to identify the risk factors for ED, statistical analysis was performed on patients diagnosed with haematological malignancies. The sample size of patients with non-haematological malignancies was too small to allow statistical analysis.

Main characteristics of patients with haematological malignancies are shown in Table 4.

Main clinical characteristics of patients in the ED group are presented in Table 5.

A statistically significant higher proportion of patients with PS 3 and 4 died within one month after diagnosis (24.1%) than patients admitted with PS 0–2 (1%)—*p* < 0.01.

Although statistical significance was not reached (*p* = 0.17), patients with leukaemia tend to have an increased risk of ED, which could be validated in a larger sample size (National cohort). We found no statistically significant difference regarding early deaths when comparing male versus female (*p* = 0.85) and age at diagnosis. There was no statistically significant difference in ED between the three periods of diagnosis (*p* = 0.7). However, we found a statistically significant difference (*p* = 0.03) regarding the performance status at diagnosis in the three diagnostic periods—data shown in Figure 2. Time interval between symptoms onset and diagnosis was not significantly different between the ED group and patients who survived beyond one month in patients with haematologic malignancies.

## 4. Discussion

The definition of ED varies among studies. Pastore et al. define the day of diagnosis as “the day on which biopsy, bone marrow examination or the most relevant imaging procedure was performed” [3]. The German Childhood Cancer Registry (GCCR) defines the date of diagnosis as the “date of admission to the hospital because of this malignancy” [9,10]. The SEER (Surveillance, Epidemiology and End Results) program defines day of diagnosis as “the day when the first diagnosis by any recognized practitioner is made” [11]. Thus, deaths due to cancer that occur before diagnosis are not captured in the Pastore definition of ED, but they are included in the GCCR and SEER definition. Taking into account that a large proportion of ED may occur before diagnostic procedures, a proportion of ED are lost when using the Pastore et al. definition.

Most studies looking into ED are population-based studies and report different results depending on definition used for ED and the source of information. A population-based study performed in Germany and published in 2020 reported 1.4% ED between 1990 and 1999 and 0.7% between 2000 and 2009 [5]. The Argentinian Oncopaediatric Registry (ROHA), mainly obtaining information from 49 oncologic paediatric units, reported 9–12% of early deaths between 2000 and 2008 [12]. Green et al. reported a percentage of 1.5% ED between 1992 and 2011 from data collected by the SEER program [4].

The rationale behind using the Pastore et al. definition of ED is that our analysis was a single institution study, the majority of patients were referred to our department by other hospitals and the first medical consult is not always recorded in patients’ files. Thus, deaths occurring before diagnosis and admission to our department were lost in the analysis. Furthermore, patients whose parents refused treatment were not captured in this analysis. Patients with CNS tumours were referred to our department only after surgical treatment. As a result, 2.55% of early deaths is an underestimation of ED in Romania. However, these results may prove important as a baseline for further investigations of ED, as a National Cancer Registry for Paediatric Cancer was founded only in 2009.

In 1996–2008, a mean of 93 new cases/year (min 81 patients, max 110 patients) were admitted in our centre. Unfortunately, a direct comparison with a nationwide case number in the same period is not possible due to the fact that the National Cancer Registry for Paediatric Cancer was founded only in 2009. However, the data from the Registry between 2010 and 2017 show a mean number of new cases of 400/year. Thus, our centre treats annually almost one-fourth of new cases diagnosed in our country.

Out of 396 patients treated in our department with solid tumours, we recorded 4 deaths (1%). Risk factors associated with ED in patients with solid tumours established in large population-based studies are age younger than one year and certain types of cancer—hepatic tumours and neuroblastoma [3,4,13]. We recorded no ED in the neuroblastoma group (representing 11% of patients with solid tumours) or in patients younger than 1 year of age (representing 5.8% of patients with solid tumours). One death was recorded in the age group 5–9, and three deaths were reported in the 1–4 years age group. All patients who died less than 30 days after diagnosis had advanced disease affecting vital organs and PS 3 or 4 at admission in our department.

Although a large number of patients with CNS tumours were treated in our department (122 patients), we recorded no ED, probably due to selection bias—most patients with CNS tumour are referred to our department after surgical treatment. Other studies have reported up to 10.8% EDs in the CNS tumour group [3].

Because of the low number of EDs in the solid tumour and CNS group, we performed statistical analysis only on patients with haematologic malignancies.

The risk factor associated with ED in patients with haematologic malignancies was PS at admission (*p* < 0.01). We found no statistical significance related to the following parameters: age at diagnosis, sex, year of diagnosis, type of haematologic malignancy and time interval between symptom onset and diagnosis.

Causes of ED in leukaemia patients are similar to those described in the literature: bleeding, mainly brain haemorrhage and leukostasis [3,14].

All EDs in lymphoma patients were recorded in patients diagnosed with non-Hodgkin lymphoma and were determined by tumour progression or complications of the disease (septicaemia, bowel obstruction). We report no EDs due to adverse events to chemotherapy or due to tumour lysis syndrome in this subgroup.

We found no difference in ED in the three defined periods of diagnosis in patients with haematologic malignancies (1996–1999; 2000–2003; 2004–2008) as we expected due to improvement in diagnostic procedures and treatment. Thus, we analysed the performance status at admission in our department and found a statistically significant increased percentage of patients with PS 3–4 in 1996–1999 versus 31.18% in 2004–2008 (*p* = 0.03).

According to our data, patients with poor performance status (PS 3–4) and patients with haematological malignancies, especially those diagnosed with leukaemia, have a higher risk of ED. Patients with high tumour volume involving or compressing vital organs should be carefully monitored and aggressively treated, as they are prone to succumb due to organ failure. We believe that early treatment with adequate supportive care is important to decrease the risk of ED in all patients, regardless of the diagnosis.

All efforts should be made to improve early access to healthcare so that fewer patients reach advanced disease and poor performance status. This can be achieved by educating the population to seek medical help for children as soon as symptoms are present, increasing the awareness of primary care physicians about childhood cancer and early diagnosis and referral of suspected cases to specialized centres.

## 5. Conclusions

As a referral centre in childhood cancer in our country, ED in our institution reflects the frequent late presentation for medical care, late diagnosis and referral to specialised centres. The increasing percentage of patients admitted in our department with PS 3–4 is of great importance, indicating that due to a variety of factors (cultural, administrative, healthcare, other unspecified issues), many patients seek medical help only when clinical status has deteriorated. Enhancing access to health services, as well as increasing trust in the national healthcare system, represents in our opinion a priority for improving treatment outcomes.

## Figures and Tables

**Figure 1 children-08-00814-f001:**
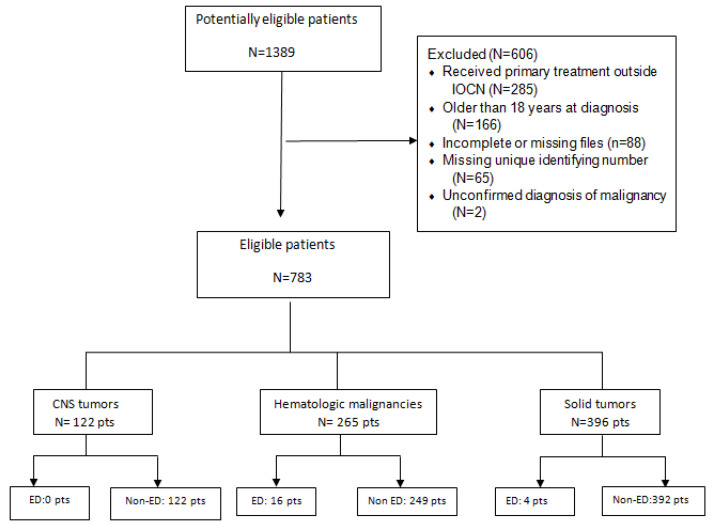
Diagram of patient flow. CNS—central nervous system; pts—patients; ED—early death.

**Figure 2 children-08-00814-f002:**
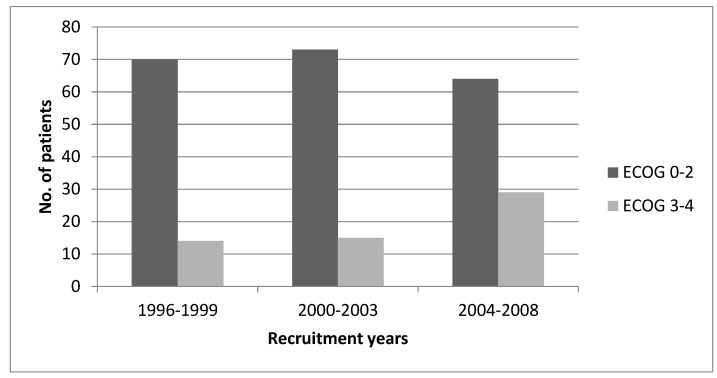
Performance status in patients with haematologic malignancies at diagnosis in 3 periods of time.

**Table 1 children-08-00814-t001:** Clinical and demographic characteristics of children with cancer who died within one month after diagnosis and of those surviving beyond that date.

Characteristic		Children Dead within One Month	Children Surviving beyond One Month	All Children *N* = 783
Sex				
Male		12 (3%)	391 (97%)	403 (51.46%)
Female		8 (2.2%)	372 (97.8%)	380 (48.54%)
Age at diagnosis (years)				
<1		-	26 (100%)	26 (3.32%)
1–4		9 (4.5%)	191 (95.5%)	200 (25.54%)
5–9		4 (2.3%)	171 (97.7%)	175 (22.34%)
10–14		7 (3.5%)	200 (96.5%)	207 (26.44%)
15–18		-	175 (100%)	175 (22.34%)
Period of diagnosis				
1996–1999		5 (2.5%)	198 (97.5%)	203 (25.92%)
2000–2004		4 (1.5%)	263 (98.5%)	267 (34.1%)
2005–2008		11(3.5%)	302 (96.5%)	313 (40%)
Diagnostic group				
CNS tumours		-	122 (100%)	122 (15.58%)
Leukaemias		11 (8%)	125 (92%)	136 (17.3%)
Lymphomas		5 (4%)	124 (96%)	129 (16.5%)
Soft tissue and other extraosseous sarcomas	Rhabdomyosarcoma	1 (2.9%)	34 (97.1%)	35 (4.4%)
	Other STS	-	29 (100%)	29 (3.7)
Extracranial GCT		1 (2.5%)	39 (97.5%)	40 (5.1%)
Renal tumours	Wilms’tumour	2 (3.6%)	54 (96.4%)	56 (7.2%)
	Other renal tumours	-	4 (100%)	4 (0.5%)
Neuroblastoma		-	44 (100%)	44 (5.6%)
Malignant bone tumours	Osteosarcoma	-	37 (100%)	37 (4.7%)
	Ewing sarcoma	-	32 (100%)	32 (4.1%)
	Chondrosarcoma	-	5 (100%)	5 (0.6%)
Malignant epithelial neoplasms and malignant melanomas	Nasopharyngeal carcinoma	-	22 (100%)	22 (2.8%)
	Thyroid carcinoma	-	40 (100%)	40 (5.1%)
	Malignant melanoma		7 (100%)	7 (0.9%)
	Other carcinoma	-	28 (100%)	28 (3.5%)
Retinoblastoma		-	15 (100%)	15 (1.9%)
Hepatic tumours		-	2 (100%)	2 (0.25%)
Performance status at admission				
0–2		2 (0.4%)	518 (99.6%)	520 (66.4%)
3–4		18 (12.6%)	124 (88.4%)	142 (18.1%)
Unknown		-	121	121 (15.5%)

STS—Soft tissue sarcoma; GCT—Germ cell tumor; “-”—no patients.

**Table 2 children-08-00814-t002:** Mortality percentage by tumour type.

Diagnostic Group	Diagnosis	Children Dead within One Month	Children Surviving beyond One Month	All Children *N* = 783
CNS tumours	Medulloblastoma	-	45	45 (5.7%)
	Ependymoma	-	13	13 (1.7%)
	Astrocytoma	-	44	44 (5.6%)
	Other CNS	-	20	20 (2.5%)
Leukaemias	ALL	6 (5.6%)	101	107 (13.7%)
	AML	5 (17.2%)	24	29 (3.7%)
Lymphomas	NHL—Non Burkitt’s	3 (8.3%)	36	39 (4.6%)
	Burkitt’s lymphoma	2 (9.1%)	23	25 (2.8%)
	Hodgkin lymphoma	-	65	65 (7.9%)
Soft tissue and other extraosseous sarcomas	Rhabdomyosarcoma	1 (2.9%)	34	35 (4.5%)
	Fibrosarcomas, peripheral nerve sheath tumours	-	3	3 (0.38%)
	Other STS	-	26	26 (3.3%)
Germ cell tumours, trophoblastic tumours and neoplasms of gonads	Extracranial GCT	1 (2.5%)	39	40 (5.1%)
	Gonadal carcinoma		4	4 (0.5%)
Renal tumour	Wilms’tumour	2 (3.6%)	54	56 (7.2%)
	Rhabdoid renal tumour	-	1	1 (0.12%)
	Kidney sarcoma	-	1	1 (0.12%)
	Renal carcinoma	-	2	2 (0.25%)
Neuroblastoma	Neuroblastoma	-	44	44 (5.6%)
Malignant bone tumours	Osteosarcoma	-	37	37 (4.7%)
	Ewing sarcoma	-	32	32 (4.1%)
	Chondrosarcoma	-	5	5 (0.6%)
Other malignant epithelial neoplasm and malignant melanomas	Nasopharyngeal carcinoma	-	22	22 (2.8%)
	Thyroid carcinoma	-	40	40 (5.1%)
	Malignant Melanoma	-	7	7 (0.9%)
	Adrenocortical carcinomas	-	1	1 (0.12%)
	Skin carcinoma	-	1	1 (0.12%)
	Other carcinomas	-	22	22 (2.8%)
Retinoblastoma	Retinoblastoma	-	15	15 (1.9%)
Hepatic tumours	Hepatoblastoma	-	1	1 (0.12%)
	Hepatic carcinoma	-	1	1 (0.12%)

ALL—acute lymphoblastic leukemia; AML—acute myeloid leukemia; STS—Soft tissue sarcoma; GCT—Germ cell tumor; CNS—central nervous system; “-”—no patients.

**Table 3 children-08-00814-t003:** Causes of death by pathophysiology among major cancer diagnostic groups.

Cause of Death	Major Diagnostic Group		
	Leukaemia	Lymphomas	Solid Tumour
Bleeding	7 pts	-	-
Infection	1 pts	-	-
Leukostasis	3 pts	-	-
Treatment related	-	-	1 pts
Organ failure due to organ compression/involvement by tumour	-	5 pts	3 pts

Pts—patients; “-”—no patients.

**Table 4 children-08-00814-t004:** Characteristics of patients diagnosed with haematological malignancies.

Characteristic.		Children Dead within One Month	Children Surviving beyond One Month	All Children	*p*
Sex	F	7 (6,4%)	103 (93.6%)	110	0.85
	M	9 (5.8%)	416 (94.2%)	155	
PS at start of treatment	0–2	2 (1%)	205 (99%)	207	<0.01
	3–4	14 (24.1%)	44 (75.9%)	58	
Year of diagnosis	1996–1999	5 (6%)	79 (94%)	84	0.70
	2000–2003	4 (4.5%)	84 (95.5%)	88	
	2004–2008	7 (7.5%)	86 (92.5%)	93	
Leukaemia	Yes	11 (8%)	127 (92%)	138	0.17
	No	5 (3.9%)	122 (96.1%)	127	
Age at diagnosis (years)	<1 year	-	3 (100%)	3	
	1–4 years	6 (8%)	69 (92%)	75	
	5–9 years	3 (4.1%)	71 (95.9%)	74	
	10–14 years	6 (8.8%)	62 (91.2%)	68	
	15–18 years	1 (2.2%)	44 (97.8%)	45	
Time interval between symptoms onset and diagnosis	<49 days	4 (4.1%)	94 (95.9%)	98	0.99
	≥49 days	5 (5.1%)	94 (94.9%)	99	
	Unknown	7 (10.3%)	61 (89.7%)	68	

**Table 5 children-08-00814-t005:** Clinical characteristics of patients included in the early death group.

Case No	Age at Diagnosis	Sex	Time Interval Between Symptoms Onset and Diagnosis (Days)	Diagnosis	Stage	PS	Disease Complications at Diagnosis	Treatment	Treatment Toxicity	Pathophysiology of Death	Cause of Death	Time Interval between Diagnosis and Death (Days)
1	4	M	26	Burkitt’s lymphoma	IIIB	2	Bowel obstruction	Cytoreductive chemotherapy	Grade 3 neutropenia, grade 3 anaemia	Bowel obstruction due to organ compression by tumour	Disease progression	24
2	13	M	Unknown	AML	NA	4	Grade 4 thrombocytopenia	Cytoreductive chemotherapy	-	Brain haemorrhage	Disease progression	23
3	13	F	15	LNH	IVB	3	Bowel obstruction	Corticotherapy		Bowel obstruction due to organ compression by tumour	Disease progression	12
4	10	F	29	AML	NA	3	Grade 4 anaemia, grade 4 thrombocytopenia, bleeding	Cytoreductivechemotherapy	-	Brain haemorrhage	Disease progression	13
5	6	M	Unknown	ALL	NA	4	Grade 4 thrombocytopenia, bleeding	Corticotherapy	-	Brain haemorrhage	Disease progression	29
6	9	M	Unknown	ALL	NA	3	Grade 4 anaemia, hyperleukocytosis, tumourlysis syndrome, bleeding	Cytoreductive chemotherapy	-	Leukostasis	Disease progression	26
7	13	F	Unknown	AML	NA	3	Grade 4 thrombocytopenia, bleeding, febrile neutropenia	Cytoreductive chemotherapy	-	Brain haemorrhage	Disease progression	13
8	1	F	4	ALL	NA	3	Tumour lysis syndrome, bleeding	Treatment of tumour lysis syndrome, supportive treatment	-	Leukostasis	Disease progression	6
9	14	M	82	ALL	-	2	Febrile neutropenia	Cytoreductive chemotherapy	-	Septicaemia	Disease progression	19
10	6	F	66	Wilms’tumour	III	3	-	Neoadjuvant chemotherapy according to SIOP 93,unilateral nephrectomy	Postoperative complications	Postoperative complications	Treatment related	30
11	1	M	unknown	Wilms’tumour	III	3	Respiratory distress due to high tumour volume	Neoadjuvant chemotherapy according to SIOP 93	-	Cardiac failure due to organ compression by tumour	Disease progression	5
12	2	M	Unknown	ALL	NA	3	Tumour lysis syndrome, hyperleukocytosis	Supportive treatment, treatment for tumour lysis syndrome	-	Leukostasis	Disease progression	4
13	4	M	Unknown	AML	NA	3	Grad 4 thrombocytopenia, bleeding	Supportive treatment	-	Gastrointestinal bleeding	Disease progression	2
14	4	M	Unknown	Burkitt’s lymphoma	IV	4	Increased liver enzymes, grade 4 thrombocytopenia	Cytoreductive chemotherapy according to NHL-BFM-95	-	Acute liver failure due to organ involvement by tumour	Disease progression	7
15	5	F	8	Large B-cell Lymphoma	IV	3	Increased liver enzymes	Cytoreductive chemotherapy	-	Acute liver failure due to organ involvement by tumour	Disease progression	24
16	2	M	Unknown	Parapharyngeal embryonal rhabdomyosarcoma	III	4	Respiratory distress	Chemotherapy according to CESS 91	-	Acute respiratory failure due to organ compression by tumour	Disease progression	5
17	14	M	Unknown	Non-Hodgkin lymphoma non-Burkitt	IVB	4	Respiratory distress	Cytoreductive chemotherapy according to NHL-BFM-95	-	Acute respiratory failure due to organ compression by tumour	Disease progression	4
18	1	F	Unknown	ALL	NA	4	Grade 4 thrombocytopenia	Cytoreductive chemotherapy according to BFM 95	-	Brain haemorrhage	Disease progression	4
19	1	M	4	Sacrococcygeal Yolk sac tumour	NA	3	Respiratory distress	Supportive treatment	-	Acute respiratory failure due to organ compression by tumour	Disease progression	12
20	12	F	Unknown	AML	NA	3	Febrile neutropenia, grade 4 thrombocytopenia	Cytoreductive and induction chemotherapy according to BFM-93	-	Brain haemorrhage	Disease progression	15

M—Male; F—Female, NA—Not applicable; PS—performance status; ALL—acute lymphoblastic leukemia; AML—acute myeloid leukemia.

## Data Availability

The date is available in the database “RISCOPED” of the Prof. Dr. Ion Chiricuta Oncology Institute.

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
