# Peer review of "Early Deaths in Childhood Cancer in Romania—A Single Institution Study"

_children, 2021, doi:10.3390/children8090814_

Round 1
Reviewer 1 Report
Early deaths in childhood cancer patients treated in the hospital is a sad experience for pediatric oncology practitioners and programs. The causes of early deaths may be complex, yet thorough descriptions and analysis of the cases can help us to learn from the failures and gain experiences to help future patients.
The authors looked into the characteristics, distribution, and causes of early deaths in a large cohort of hospitalized children with cancer. The good news is that the early death rate remains low at 2.55%, despite the higher proportion of patients with low performance status. However, the reporting of cancer classification and analysis of cause of deaths require revisions before publication. (see below)
Please describe the service volume of the authors' institution. If possible, please describe the proportion of annual new patients treated at the institution vs. nationwide case numbers.
Materials and Methods: Please provide a CONSORT diagram of patient flow.
Please confirm if a research ethics review was required and performed for this work.
Line 69: Please specify the statistical test used to compare categorical data.
Line 92: Causes of early death: Were there early deaths that were related with treatment abandonment?
Can the authors further reason the causes of early deaths according to the classification of causes of treatment failure established by SIOP PODC? (i.e. Relapse, abandonment, misdiagnosis, toxic death, no treatment, no diagnosis)
The following references may be helpful:
Lam et al., Science 2019 https://science.sciencemag.org/content/363/6432/1182
Howard et al., Lancet Oncol 2018
https://pubmed.ncbi.nlm.nih.gov/29726390/
The causes of early death seem to be segregated with cancer types - more bleedings in leukemias and more organ failures in solid tumors. The authors can also try to classify the causes of death by pathophysiology and compare among major cancer diagnostic groups.
The authors should further discuss the potential strategies to further prevent early deaths. How would the authors treat future patients with ECOG 3/4 to prevent early deaths? Should leukemia and solid tumor patients use the same strategy to prevent early deaths? Some "risk stratification" may be proposed.
Table 1. Diagnostic group and Table 2 - The diagnostic classification will be more clear to be listed according to the International of Childhood Cancer Classifications version 3 (ICCC-3) major and minor groups. Some smaller groups can be grouped/collapsed together in Table 1, e.g. embryonal tumors, sarcomas, others and carcinomas.
Table 3. Leukemia - Yes vs. No: This variable seems to have a trend
Tables 1 and 3 look redundant. May the authors compare the categorical data in Table 1 directly?
The authors may also make a Table or Supplemental Table to describe the clinical characteristics; cause of treatment failure; pathophysiology of death; and description of complications/deaths in the 20 patients with early deaths. Such a table may be easier to get summary information of the early deaths.
Author Response
Thank you for giving us the opportunity to submit a revised draft of our manuscript. We appreciate the time and effort that you have dedicated to providing your feedback on our manuscript. We incorporated changes that reflect most of the suggestions provided. Changes have been highlighted in the manuscript.
Materials and Methods: Please provide a CONSORT diagram of patient flow.
- We added Consort diagram: Figure 1.
Please confirm if a research ethics review was required and performed for this work.
The study was reviewed and approved by the Ethics Committee of Iuliu Hațieganu University of Medicine and Pharmacy. We inserted this information in the text
Line 69: Please specify the statistical test used to compare categorical data.
- Comparison of categorical variables was performed using the chi-square (χ2) test. P < 0.05 was considered statistically significant.
Line 92: Causes of early death: Were there early deaths that were related with treatment abandonment?
- We added the following information: treatment abandonment defined according to SIOP PODC was registered in 58 patients (Seven patients refused treatment and 51 patients abandoned treatment), however no early deaths were reported in this group of patients
Can the authors further reason the causes of early deaths according to the classification of causes of treatment failure established by SIOP PODC? (i.e. Relapse, abandonment, misdiagnosis, toxic death, no treatment, no diagnosis)
The following references may be helpful:
Lam et al., Science 2019 https://science.sciencemag.org/content/363/6432/1182
Howard et al., Lancet Oncol 2018
https://pubmed.ncbi.nlm.nih.gov/29726390/
- We added the following information in text: From the 20 ED registered in this period, 19 deaths were related to disease progression, and one death was related to treatment (surgery) (see table 4). No ED related to relapse, misdiagnosis or failure to establish diagnosis were recorded in this cohort of patients.
The causes of early death seem to be segregated with cancer types - more bleedings in leukemias and more organ failures in solid tumors. The authors can also try to classify the causes of death by pathophysiology and compare among major cancer diagnostic groups.
- We have included Table 3 in the manuscript to provide this information.
The authors should further discuss the potential strategies to further prevent early deaths. How would the authors treat future patients with ECOG 3/4 to prevent early deaths? Should leukemia and solid tumor patients use the same strategy to prevent early deaths? Some "risk stratification" may be proposed.
- We have added potential strategies in the Discussion section.
Table 1. Diagnostic group and Table 2 - The diagnostic classification will be more clear to be listed according to the International of Childhood Cancer Classifications version 3 (ICCC-3) major and minor groups. Some smaller groups can be grouped/collapsed together in Table 1, e.g. embryonal tumors, sarcomas, others and carcinomas.
- Thank you for this suggestion. We have modified both tables according to ICC-3 groups in the manuscript
Table 3. Leukemia - Yes vs. No: This variable seems to have a trend
- In order to underline this we added the following text: Although statistical significance was not reached (p.0.17), patients with leukemia tend to have an increased risk of ED, this could be validated in a larger sample size (National cohort).
Tables 1 and 3 look redundant. May the authors compare the categorical data in Table 1 directly?
- Table 1 shows clinical and demographic characteristics in all children included in this analysis, while table 3 shows a statistical analysis performed only on patients diagnosed with hematological malignancies. For this reason, we would rather keep these two tables separate.
The authors may also make a Table or Supplemental Table to describe the clinical characteristics; cause of treatment failure; pathophysiology of death; and description of complications/deaths in the 20 patients with early deaths. Such a table may be easier to get summary information of the early deaths.
- Thank you for this suggestion. We added this information in table 5
Reviewer 2 Report
The authors present a retrospective analysis of children patients (aged 0-18 years) with cancer diagnosed between 1996 and 2008 and admitted in the Institute of Oncology “Prof. Dr. Ion Chiricuta” Cluj-Napoca (IOCN) in Romania. The institutional analysis scoped on early death in this population. The topic is interesting.
I have several questions:
-was there any relation of ED and cancer stage at the time diagnosis?
-was all cases autopsied? If not why? Was cause of death cancer related in all cases?
Some tables can be replaced by graphs.
Author Response
Thank you for giving us the opportunity to submit a revised draft of our manuscript. We appreciate the time and effort that you have dedicated to providing your feedback on our manuscript. We incorporated changes that reflect most of the suggestions provided. Changes have been highlighted in the manuscript.
- was there any relation of ED and cancer stage at the time diagnosis?
Due to the low number of early deaths registered in the solid tumor group of patients, we could not perform a statistical analysis regarding the stages. A larger cohort is needed in order to validate the assumption that early deaths are more frequent in patients with higher stages.
- was all cases autopsied? If not why? Was cause of death cancer related in all cases?
No patients in the early death group were autopsied. In our country parents have the right to refuse autopsy when the major cause of death (in our case cancer) is known. All parents of patients included in this analysis refused autopsy.
- Some tables can be replaced by graphs
Thank you for this suggestion. We were able to replace Table 4 with a graph.
Round 2
Reviewer 1 Report
The authors have incorporated previous suggestions into the revised version. Three minor comments:
- In Tables 1,2,3, Column 1, Children "dead" within one month --> Using the word "died" seems more appropriate here.
- Bleeding is the major cause of ED in this cohort (n = 7/20). The current discussion about future strategies mainly focused on organ compression (Lines 240-242). Can the authors further comment on how to prevent EDs due to bleeding in the future? (e.g. Blood banking strategies; transfusion guidelines; promotion for blood donation; or social awareness campaign or other efforts to decrease late presentation)
- Lines 108-111 and Figure 1: Although abandonment to treatment is rare in Europe, can the authors confirm that there is no treatment abandonment? Especially, were there patients who were abandoned to treatment (missing their scheduled treatment for more than 1 month) excluded from the analysis?
Author Response
Thank you for your time and effort.
We have made the changes suggested. We have highlighted the changes within the manuscript.
In Tables 1,2,3, Column 1, Children "dead" within one month --> Using the word "died" seems more appropriate here.
- We have modified the tables as suggested
- Bleeding is the major cause of ED in this cohort (n = 7/20). The current discussion about future strategies mainly focused on organ compression (Lines 240-242). Can the authors further comment on how to prevent EDs due to bleeding in the future? (e.g. Blood banking strategies; transfusion guidelines; promotion for blood donation; or social awareness campaign or other efforts to decrease late presentation)
- We have inserted the suggested information in text.
Immediate access to necessary quantities of blood products, especially platelets, is an important supportive care measure in patients with hematologic malignancies, however this is not always possible due to low number of donators. Increasing awareness for blood donation in the general population could indirectly decrease the number of ED in this group of patients.
- Lines 108-111 and Figure 1: Although abandonment to treatment is rare in Europe, can the authors confirm that there is no treatment abandonment? Especially, were there patients who were abandoned to treatment (missing their scheduled treatment for more than 1 month) excluded from the analysis?
We have clarified information in text. Patients with treatment abandonment were included in our analysis.